# Role of CAR T Cell Metabolism for Therapeutic Efficacy

**DOI:** 10.3390/cancers14215442

**Published:** 2022-11-04

**Authors:** Judit Rial Saborido, Simon Völkl, Michael Aigner, Andreas Mackensen, Dimitrios Mougiakakos

**Affiliations:** 1Department of Internal Medicine 5, Hematology and Oncology, Friedrich-Alexander-Universität and University Hospital Erlangen, 91054 Erlangen, Germany; 2Deutsches Zentrum für Immuntherapie (DZI), Friedrich-Alexander-Universität and University Hospital Erlangen, 91054 Erlangen, Germany; 3Medical Center, Department of Hematology and Oncology, Otto-von-Guericke University, 39120 Magdeburg, Germany

**Keywords:** CAR T cells, metabolism, bioenergetics, immune escape, immunometabolism, tumor microenvironment

## Abstract

**Simple Summary:**

Chimeric antigen receptor (CAR) T cell therapy has heralded a new era in cancer treatment, in particular for hematological malignancies. Despite the current progress in CAR T cell research and development, frequent occurrence of exhausted and/or terminally differentiated CAR T cells can lead to poor tumor infiltration, limited persistence, lack of effector functions, and finally tumor immune escape. In fact, key functions and even the differentiation of T cells are tightly interconnected with the cells’ bioenergetics. Tumor cells and their microenvironment (TME) in turn can impact T cell metabolism in a variety of ways, including depletion of critical nutrients (e.g., glucose or tryptophan), accumulation of bioactive metabolites (e.g., lactic acid or reactive oxygen species) or via immunological checkpoints. Given this strong link between T cell metabolism and functional features that represent prerequisites for an efficient CAR T cell therapy, it is of great interest to explore metabolic modulation as a mean to improve clinical efficacy and even tolerability.

**Abstract:**

Chimeric antigen receptor (CAR) T cells hold enormous potential. However, a substantial proportion of patients receiving CAR T cells will not reach long-term full remission. One of the causes lies in their premature exhaustion, which also includes a metabolic anergy of adoptively transferred CAR T cells. T cell phenotypes that have been shown to be particularly well suited for CAR T cell therapy display certain metabolic characteristics; whereas T-stem cell memory (T_SCM_) cells, characterized by self-renewal and persistence, preferentially meet their energetic demands through oxidative phosphorylation (OXPHOS), effector T cells (T_EFF_) rely on glycolysis to support their cytotoxic function. Various parameters of CAR T cell design and manufacture co-determine the metabolic profile of the final cell product. A co-stimulatory 4-1BB domain promotes OXPHOS and formation of central memory T cells (T_CM_), while T cells expressing CARs with CD28 domains predominantly utilize aerobic glycolysis and differentiate into effector memory T cells (T_EM_). Therefore, modification of CAR co-stimulation represents one of the many strategies currently being investigated for improving CAR T cells’ metabolic fitness and survivability within a hostile tumor microenvironment (TME). In this review, we will focus on the role of CAR T cell metabolism in therapeutic efficacy together with potential targets of intervention.

## 1. Introduction

Since its first clinical application in 2011, CAR-T cells have revolutionized the field of cancer therapy. As outcome, several approvals by the Food and Drug Administration (FDA) have followed [1]. CARs are synthetic constructs that bind to a specific target antigen in a major histocompatibility complex (MHC)-independent fashion. MHC binding triggers a vigorous T cell activation cascade leading to the target cells’ elimination. The currently approved CAR T cell constructs consist of four components: an extracellular target antigen-binding domain, a hinge region, a transmembrane domain, and one or more intracellular signaling domains [2,3,4]. In fact, co-stimulation not only acts as an enhancer of T cell signaling and promotion of a memory-like phenotype, it also controls the CAR T cells’ metabolic phenotype, which is pivotal for function, differentiation, and longevity [5]. T cells that carry CARs with a 4-1BB co-stimulatory domain have higher OXPHOS rates and an increased respiratory reserve, while cells with CD28 are glycolytic, which could both be beneficial in the respective scenarios of a glucose-depleted and/or hypoxic TME. The third generation CARs even incorporate a second co-stimulatory signaling domain to foster T cell viability, proliferation, and effector functions [6]. Moreover, numerous additional approaches are being pursued to optimize CAR T cell treatment. These include the combination with other therapeutic modalities (e.g., immune checkpoint blockade [7,8,9] or immunomodulatory drugs (IMIDs) [10,11,12]), the use of “off-the-shelf” allogeneic CAR T cells [13,14] as well as the co-targeting of multiple antigens (e.g., CD19xCD20 or CD19xCD22) [15,16]. 

## 2. Current Limitations of CAR T Cell Therapy

Despite the aforementioned progress, CAR T cell therapy still faces several obstacles, including treatment-related toxicity, limited efficacy against solid tumors, loss of target antigen, poor tumor infiltration, and limited persistence, driven, in part, by the immunosuppressive TME (as summarized in Figure 1). 

### 2.1. CAR T Cell-Associated Toxicities

Several CAR-related (i.e., target antigen, co-stimulatory domain) and disease-related (i.e., type of tumor, tumor mass, pro-inflammatory activity) factors determine the incidence and extent of therapy-associated toxicities [3]. The most relevant include cytokine-release syndrome (CRS) [4], immune effector cell-associated neurotoxicity syndrome (ICANS) [2,4], and long-lasting hematotoxicity [17]. CRS results from extensive T cell activation and massive cytokine release, mostly IL-6, by myeloid cells. Therefore, IL-6 receptor blockade by tocilizumab represents the first line treatment [18]. The pathophysiology of ICANS is not fully elucidated yet. Unlike CRS, IL-6 does not play a prominent role and first line treatment consists of steroids. The situation is similar for hematotoxicity, where therapy is symptomatic with, e.g., transfusions, autologous stem cell boost, and use of thrombopoietin receptor agonists [19,20,21]. For instance, it has been shown that CD28 CAR T cells induce more severe adverse effects than 4-1BB CAR T cells, including higher frequency of grade III-IV CRS, grade I-II neurotoxicity and episodes of severe ICANS [22].

### 2.2. Poor Trafficking and Infiltration in Solid Tumors

Treatment of solid malignancies is decisively dependent on the CAR T cells’ ability to infiltrate the tumor site. Additionally, clinical efficacy has not been nearly as good as for hematological malignancies. Strategies to overcome this obstacle include local CAR T cell administration [23,24], armament of CAR T cells with chemokine receptors responsive to tumor-derived chemokines [25], and expression of enzymes (e.g., heparinase) that degrade the extracellular matrix [26]. Interestingly, T cell metabolism can influence cell motility and consequently tumor tissue infiltration. T cell motility mainly relies on amoeboid-like motion [27]. Naturally, both glycolysis and OXPHOS fuel this energy-consuming process [28,29]. In addition, depletion of certain nutrients such as tryptophan and arginine [30,31] or the accumulation of lactic acid [29] within the TME can inhibit T cell motility.

### 2.3. Loss of Target Antigen

One of the best-described immune escape mechanisms is the loss of the target antigen because of the immunological pressure. Countermeasures include the development of multi-specific CARs (e.g., dual or tandem CARs) that target more than one tumor antigen. Signals from clinical trials are encouraging [20,21].

### 2.4. Immunosuppressive TME

The TME is a hostile milieu for (CAR) T cells, also from an immunometabolic perspective. Nutrients required for proper T cell function, such as glucose or arginine, are depleted, whereas detrimental metabolites such as kynurenine, reactive oxygen species (ROS), and lactic acid accumulate at the same time. The expression of immune checkpoint molecules such as programmed cell death ligand 1 (PD-L1) interfere with T cell function and bioenergetics [32]. Moreover, the presence of tolerance-promoting cell subsets such as myeloid-derived suppressor cells (MDSCs), tumor-associated macrophages (TAMs) or regulatory T cells (T_Regs_) antagonizes tumor-directed immune responses [33]. Interestingly, metabolic crosstalk can be part of this process. For instance, MDSCs paralyze T cells by transferring the metabolite methylglyoxal, which is a byproduct of MDSCs metabolism that was found to be enriched in patient-derived MDSCs. After co-culture of T cells and MDSCs, it was found that T cells displayed enhanced concentrations of this metabolite, resulting in impaired glycolysis, cell proliferation and T cell function, as evidenced by reduced cytokine production [34]. On the other hand, T_Regs_ inhibit T cell function in the TME via several mechanisms. First, T_Regs_ deprive effector T cells from IL-2 given their high expression of IL-2R. In addition, they secrete inhibitory cytokines, including TGF-β, IL-10 and IL-35, and are involved in immune checkpoint-related pathways, such as the interaction between PD-1/PD-L1 or the impairment of antigen presentation by downregulation of CD80/86 expression caused by CTLA-4 expressed on T_Regs_ [35]. In relation to metabolism, T_Regs_ modulate the REDOX balance, convert ATP into adenosine and induce the expression of indoleamine 2,3-dioxygenase (IDO) in dendritic cells, which leads to T cell exhaustion via essential amino acid depletion [35,36]. All of the aforementioned factors can attenuate T cell-based immune responses and lead to (premature) exhaustion.

## 3. Link between T Cell Differentiation and Metabolism

As mentioned above, therapeutic efficacy and persistence of CAR T cells significantly correlate with their differentiation state (Figure 2). The different stages of T cell differentiation include the following: naïve T cells (T_N_), stem central memory T cells (T_SCM_), central memory T cells (T_CM_), effector memory T cells (T_EM_), effector T cells (T_EFF_), and terminally differentiated T cells (T_EMRA_). CAR T cell products with a high content of T_N_, T_CM_, and T_SCM_ cells display a superior antitumor response and in vivo persistence [37,38]. This observation is most likely because less differentiated T cell subsets (such as T_N_, T_CM_, and T_SCM_) are characterized by an enhanced capacity for self-renewal [39]. The differentiation status of T cells and metabolic phenotypes are tightly interconnected and since adoptive immune cell therapies appear to benefit from products rich in less differentiated CAR T cells, it is important to understand how metabolism shapes differentiation and vice versa. 

As T_N_ leave the thymus, they mostly rely on OXPHOS fueled by glucose-derived pyruvate or by fatty acid oxidation (FAO) [40,41,42]. Once in the periphery, T_N_ are likely to encounter antigens, to undergo T cell receptor (TCR) activation, and differentiation towards T_EFF_ cells. This process is paralleled by metabolic skewing towards aerobic glycolysis despite the presence of sufficient oxygen for performing OXPHOS [43,44]. This metabolic reprogramming allows production of metabolic intermediates required for rapid cell growth and proliferation (upon stimulation), and the maintenance of the cellular redox balance during stress conditions. This process is tightly regulated by the key metabolic regulator, mammalian target of rapamycin (mTOR) [45,46,47]. In addition to glycolysis, it has been proven that mitochondrial activity is also important to support T cell effector function. Additionally, this is via the generation of reactive oxygen species (ROS) in the electron transport chain (ETC) since ROS have been shown to be essential to maintain the T cell activation-induced metabolic shift and support T cell proliferation and IL-2 production [48,49,50]. 

Antigen encounter and its successful clearance are followed by a contraction of the T_EFF_ cell population. However, a small subset persists as long-lived memory T cells, characterized by rapid activation, proliferation, and effector function upon antigen re-exposure [51]. In contrast to T_EFF_ cells, mTOR inhibition via rapamycin enhances memory T cell formation in vivo [52]. In this case, 5′ adenosine monophosphate-activated protein kinase (AMPK) regulates memory T cell metabolism [53]. Memory T cells fuel their energetic demands through FAO (and OXPHOS). This leads to increased spare respiratory capacity (SRC), which allows T cells to rapidly meet the energetic demands in situations of stress or nutrient restriction, such as those encountered during an infection or within the TME [53,54]. Within the memory subset, there are slight differences between T_CM_ and T_EM_. Whereas T_CM_ strongly rely on FAO and OXPHOS to maintain their robust proliferative response and cytokine production upon re-stimulation, T_EM_ tend to be less metabolically dependent on OXPHOS [55]. Similar to T_CM_ and T_EM_, T_SCM_ also display reduced glucose metabolism with a preference for lipid metabolism, characterized by high SRC. However, T_SCM_ are characterized by a lower mitochondrial membrane potential [56].

Different T cell subsets rely on distinct metabolic programs that determine the function, longevity, persistence, and consequently success of T cell-based immunotherapies. Several clinical T cell trials report a positive correlation between the infusion of less differentiated T cells and better clinical outcome. For instance, the percentage of T_N_ or T_CM_ in the infused products has been positively correlated with the prolonged presence of transferred T cells in the peripheral blood and with an overall response in different malignancies, including melanoma [57,58,59], neuroblastoma [60], chronic lymphocytic leukemia [61], multiple myeloma [62], pancreatic carcinoma [63] and B cell lymphoma [64]. Strategies to promote stemness include, amongst others, application of cytokines (e.g., IL-7 or IL-15), activation of certain signaling pathways (e.g., Wnt and Notch) or epigenetic modulation (e.g., bromodomain inhibitors). In line with these observations, CD19 CAR T cells derived from T_SCM_ cells yield better long-term responses as compared to their T_EFF_ counterparts [37].

## 4. T Cell Metabolism in the TME

T cell metabolism is not only impacted by the state of T cell differentiation and activation, etc., but also by the environmental conditions, which can be rather burdensome in the case of the TME (Figure 3). One of the reasons is that activated T cells and malignant cells have a similar metabolic profile and thus metabolic requirements, which leads to competition over nutrients such as glucose [65]. This competition can impede T cell responses as one exemplary study showed an inverse correlation between expression of the glucose transporter 1 (Glut1) in renal cell carcinoma with T cell function and infiltration [66]. Another indirect mechanism whereby glycolytic tumors suppress T cell function is via excessive lactic acid release. Lactic acid has inhibitory effects on T cell metabolism, in particular glycolysis, which results from an inauspicious lactic acid gradient between the extracellular milieu and the cytoplasm [67,68]. In fact, numerous tumor-derived metabolic by-products hold a suppressive potential. The accumulation of potassium [K^+^] within the TME blocks amino acid and glucose transporters of T cells [69]. The ectonucleotidases CD39 and CD73, which can be found highly expressed on tumors, convert extracellular ATP into adenosine. The latter interferes with NfκB signaling, limiting the T cells’ anti-tumor activity and promoting the induction of T_Regs_ [70,71]. Adenosine also directly impacts T cell metabolism and function, which correlates with the abundance of the adenosine receptor (A2AR). Indeed, adenosine blunts mTOR target, S6, phosphorylation, which leads to a reduction in glycolytic metabolism. Adenosine also seems to reduce OXPHOS, but the effects are not as strong. In addition to impairing metabolism, adenosine reduces cytokine production and T cell degranulation activity [72]. 

Moreover, increased concentrations of lipids (e.g., cholesterol) within the TME can trigger so-called lipotoxicity leading to cell death and exhaustion of T cells [73]. In a different study it was also shown that accumulation of long-chain fatty acids in the TME induced T cell dysfunction in pancreatic cancer [74]. T cell metabolic dysfunction in the TME has been described in other studies where, for example, T cells from CLL patients show reduced glucose uptake and increased SRC and membrane potential, accompanied by increased ROS [75]. Similarly, in a model of melanoma, intratumoral T cells were characterized by depolarized mitochondria and functional exhaustion [76]. 

In addition to the aforementioned molecules, different cytokines in the TME also have an impact on T cell metabolism and differentiation. On the one hand, IL-2 sustains the glycolytic shift via sustained mTOR activation, and this has been shown to favor T_Regs_ generation and suppressive activity given the high expression of the IL-2 receptor in this subset [77,78]. On the contrary, cytokines such as IL-7, IL-15 and IL-21 are known to drive mitochondrial and fatty acid metabolism via the increase in glycerol uptake, SRC, CPT1α expression and even the expression of anti-oxidant molecules (glutathione reductase, thioredoxin reductase 1, peroxiredoxin and superoxide dismutase) that compensate for the increase in ROS following OXPHOS [79,80,81,82]. The prevalence of mitochondrial metabolism is linked to a less differentiated pool of T cells. In the TME, inflammatory cytokines, such as IFN-γ, IL-1β, IL-23 and TNF-α, are abundant. All of these cytokines contribute to the enrichment of differentiated effector T cells mainly via mTOR activation [83,84,85,86]. Conversely, anti-inflammatory cytokines, such as TGF-β, have the potential to inhibit mTOR activity, limiting glycolysis and subsequently shifting the balance towards a mitochondrial-based less differentiated phenotype [87]. To date, immunological checkpoints such as PD-L1 are well-established drivers of tumor immune escape. Recent data suggest that they exert part of their effects by interfering with T cell metabolism. The binding of PD-L1 to its cognate receptor PD-1, expressed on T cells, impairs their effector functions by reducing glycolytic activity. At the same time FAO is enhanced but PD-1^+^ T cells display an overall reduced SRC [32]. Moreover, PD1^+^ T cells also show a reduced peroxisome proliferator-activated receptor gamma coactivator 1-alpha (PGC1α) expression, which controls mitochondrial biogenesis [32]. A recent study highlighted the importance of mitochondrial biogenesis for manufacturing effective CAR T cells in chronic lymphocytic leukemia (CLL) [75]. Interestingly, PD-L1 expressed on tumor cells boosts their glucose uptake, further limiting glucose availability for infiltrating T cells [65]. The second very well-described inhibitory receptor, CTLA-4, downregulates glutamine transporters (SNAT1 and SNAT2) and Glut1 [32]. These results suggest that the immune checkpoint blockade as currently employed could restore immunometabolic fitness, which needs to be further investigated. In this direction, several clinical trials (NTC00586391, NTC01822652, NCT02650999, NCT02706402, NCT02926833, NCT03310619, NCT03726515, NCT02862028, NCT03081715, NCT02867332, NCT02867345, NCT02793856, NCT03044743) focus on the combination of CAR T cells with an immune checkpoint blockade (CPB). These studies mostly focus on the inhibition of the PD-1/PD-L1 axis via different strategies, which include the administration of anti-PD1 and PD-L1 antibodies, the design of anti-PD-1-blocker secreting CAR T cells and the genetic modification of CAR T cells to knock out the PD-1 receptor. Most of these studies show a benefit of the combination treatment in terms of T cell function and proliferation. However, they are limited by the short half-life of blocking antibodies and associated toxicities [7,8,9,88]. Therefore, although promising, these strategies need to be optimized. 

## 5. Impact of CAR Design on T Cell Metabolism

A less differentiated T cell phenotype appears advantageous for the clinical efficacy of CAR T cells. The design of certain CAR components could co-determine the metabolic phenotype together with stemness (Figure 4).

### 5.1. Co-Stimulatory Domains

Various co-stimulatory domains are used in CARs. Currently, most constructs belong to one of the following superfamilies: the tumor necrosis factor receptor superfamily (4-1BB/CD137, OX40) or the immunoglobulin superfamily (CD28, ICOS/CD278) [89]. Most of the studies that compare 4-1BB vs. CD28 involved CARs directed against CD19 or mesothelin. Kawalekar and colleagues describe a higher basal oxygen consumption rate (OCR), which is a surrogate for OXPHOS, and SRC in 4-1BB- as compared to CD28-carrying CARs [90]. This difference was strongest for the T_CM_ and T_N_ populations. On the contrary, metabolic flux analyses revealed enhanced extracellular acidification rates (ECAR), a surrogate for aerobic glycolysis, in CD28 CAR T cells. The reliance of 4-1BB CAR T cells on FAO, which represents a feature of memory-like T cells, was confirmed using heavy-carbon-labelled palmitic acid. In accordance with the superior ECAR, CD28 CAR T cells display an increased expression of key genes involved in glycolysis, including *GLUT*1, phosphoglycerate kinase (*PGK*), and glucose-6-phosphate dehydrogenase (*G6PD*). Enhanced FAO, SRC, and OXPHOS in 4-1BB CAR T cells is supported by an enhanced mitochondrial biogenesis as well as fusion, which is also seen in memory T cells. These differences might arise from the molecular pathways that are triggered by each co-stimulatory domain. For instance, CD28 interacts with PI3K leading to Akt activation, which in turn induces the expression of glucose transporters and other enzymes involved in glucose metabolism [46,91]. Alternatively, 4-1BB activates the p38-MAPK signaling axis, which results in PGC1α overexpression and subsequent mitochondrial fusion and biogenesis [92,93]. 

Those metabolic features might explain the preferential skewing towards a T_CM_ phenotype, slower effector responses, and longer in vivo persistence of 4-1BB CAR T cells. In a similar study, Liu and colleagues reported that OCR is similar for CAR T cells with both types of co-stimulatory domains but 4-1BB CARs display an enhanced SRC [94]. The introduction of 4-1BB into the CAR construct led to a downregulation of genes encoding for glycolytic molecules and an upregulation of genes (such as the fatty acid-binding protein 5) involved in FAO and mitochondrial metabolism.

Overall, the data to date indicates that 4-1BB co-stimulation skews CAR T cell metabolism towards OXPHOS and FAO and promotes a less differentiated phenotype (with a higher replicative potential). Interestingly, third generation CAR T cells, which bare both CD28 and 4-1BB co-stimulatory domains, seem to preserve the high mitochondrial metabolism of 4-1BB CAR T cells but are accompanied by increased glycolytic metabolism. This overall enhanced metabolic activity prompted a sustained tonic TCR signaling, proliferation and metabolic fitness in dual CAR T cells over time [95]. 

### 5.2. Other CAR Construct-Related Factors

In addition to the co-stimulatory domains, other aspects of CAR design can influence the metabolism of CAR T cells. Li and colleagues showed that ubiquitination-deficient CD19 CAR T cells displayed higher OCR and SRC [96]. The inhibition of ubiquitination led to a recycling of CAR constructs to the surface and enhanced lysosomal signaling, which promotes OXPHOS and, subsequently, memory T cell formation. A different study focused on the dynamics of CAR T cell stimulation, in particular in T_Regs_ with tonic CAR receptors, which uncouple antigen recognition from CAR activation signaling. Here, they reported that tonic CAR signaling led to an exhausted T cell phenotype with higher ECAR and OCR but low SRC. Moreover, T cells carrying a tonic CAR construct relied more on glycolysis in terms of energy [97]. 

## 6. Strategies to Target CAR T Cell Metabolism

With increasing data regarding the importance of energy metabolism for the function and persistence of CAR T cells, many strategies of intervention are currently being exploited, which we summarize in the follow (Figure 5).

### 6.1. Optimizing Cell Culturing Conditions

Culture conditions during CAR T cell manufacturing can determine the differentiation state and metabolic phenotype. One important factor is serum supplementation. Serum, either of animal or human origin, is an important component of media to support cell viability and growth. The traditional formulations include fetal bovine serum (FBS) and human serum (HS). Recent studies explored the usage of human platelet lysates (HPL) that are used for the expansion of mesenchymal stem cells (MSCs) [98]. CAR T cells expanded in presence of HPL (as opposed to cells cultured with FBS- or HS-containing media) displayed an enrichment of T_CM_ cells, a superior in vivo expansion, and an enhanced anti-tumor activity [99,100]. The metabolic effects of HPL supplementation remain to be investigated. As an alternative, serum can be replaced by Physiologix^TM^ (Phx), which is an extract of whole blood-derived growth factors. Phx enhanced the functionality and in vivo expansion of CAR T cells. At the same time, metabolism was skewed away from glycolysis towards OXPHOS, which is triggered by an enrichment of the dipeptide carnosine by neutralization of the extracellular H^+^ [101].

The duration of the CAR T cell production protocol can affect differentiation and function. Typically, the CAR T cell expansion lasts between 9 and 14 days. However, shorter intervals allow for the generation of CAR T cells with higher proliferative capacity, less differentiation, and greater cytotoxic activity. These optimized protocols range from three to five days [102].

The most commonly used cytokine for CAR T cell production is IL-2. However, IL-2 promotes glycolysis for rapid proliferation but leads to the formation of short-lived, highly differentiated, and exhausted CAR T cells [103,104]. Efforts to substitute IL-2 have focused on alternative γ-chain cytokines, including IL-7, IL-15, and IL-21. Evidence supports that IL-15 preserves stemness and induces higher anti-tumor activity and proliferation via mTOR inhibition. Interfering with mTOR signaling leads to reduced glycolysis and a shift towards OXPHOS [105]. Furthermore, combining IL-15 and IL-7 further boosts generations of T_SCM_ cells [106]. In addition, IL-21 represents an interesting candidate since it enhances FAO, thus supporting T_CM_ cell generation and in vivo persistence. Furthermore, IL-21 might yield a superior anti-tumor activity as compared to IL-15 [107]. Interestingly, the administration of a PD-1/IL-21 fusion protein improved the delivery of IL-21 to PD-1 expressing T cells, which was superior compared to systemic delivery [108,109]. Other strategies include the combination of IL-21 with IL-4 and IL-7, which maintained stemness and reduced the expression of inhibitory receptors, including PD-1 and TIM3 in CAR T cells [110]. 

### 6.2. Interfering with Glycolysis

The TME is often characterized by a shortage of nutrients. Malignant cells and T cells compete over bioenergetic substrates. Generally, one should assume that lack of nutrients is detrimental for (T) cells. However, recent studies suggest that metabolic stress, particularly glucose deprivation, may promote T cell persistence. One of the most studied glycolysis inhibitors is 2-deoxy-D-glucose (2-DG). 2-DG is a glucose derivative that enters cells via glucose transporters and is then phosphorylated by hexokinase 2 (HK2), the pacemaker enzyme of glycolysis. The resulting 2-DG-6-phosphate cannot be further metabolized and HK2 activity decreases significantly due to end product inhibition [111]. As anticipated, the treatment of T cells during in vitro expansion with 2-DG favored formation of memory T cells [112]. More recently, it was shown that transient glucose restriction improves the T cell anti-tumor function via increased pentose phosphate pathway (PPP) activity [113]. The PPP is a key pathway that generates the reduced form of Nicotinamide adenine dinucleotide phosphate (NADPH), pentoses and ribose-5-phosphate, which serve as antioxidants and nucleotide precursors, respectively. In the latter study, by transiently restricting glycolysis, they could show that T cells had a more reduced state and more PPP-generated intermediates to sustain their functionality. 

One of the main pathways activated upon TCR engagement (and further boosted by CD28 stimulation) is the PI3K/Akt/mTOR signaling axis, which promotes glycolysis to support rapid T cell proliferation and differentiation. Therefore, interfering with this signaling pathway could lead to less-differentiated CAR T cells. First, Perkins and colleagues tested the expansion of B cell maturation antigen (BCMA)-directed CAR T cells in the presence of a PI3K inhibitor. The infusion of these CAR T cells to Burkitt lymphoma- and multiple myeloma-bearing mice resulted in long-term tumor regression. One of the key findings was the enhanced frequency of CD8^+^ CD62L^+^ memory T cells [114]. Similarly, Petersen and colleagues targeted the delta subunit of PI3K (δPI3K), expressed specifically in lymphocytes. In this case, both murine and human cytotoxic lymphocytes (CTLs) displayed a less differentiated phenotype as compared to their untreated counterparts [115]. The effects of tonic CAR signaling that can promote glycolysis and exhaustion can also be antagonized by pharmacological PI3K inhibition as shown for CD33-specific CAR T cells [116].

In equivalence to PI3K inhibition, Akt blockade skews metabolism towards FAO and OXPHOS, which is accompanied by enhanced stemness. However, these effects were independent of a reduced glycolytic flux. Accordingly, inhibition of Akt in EpCAM-specific CAR T cells prevented terminal differentiation without having an impact on viability and proliferation [115]. Furthermore, treated CAR T cells display a superior expansion and antitumor activity in preclinical colon cancer models. Inhibition of mTOR causes similar phenomena. Treatment of IL-2-expanded CAR T cells led to reduced glycolysis and a less-differentiated phenotype [105]. It is worth noting that intervention in the PI3k/Akt/mTOR axis not only promotes the induction of memory T cells via the blockade of glycolysis, but also via increased autophagy. This process of clearing debris and damaged organelles is very important for cellular homeostasis [117]. Additional targets of the PI3K/Akt/mTOR axis with potential (direct) impact on stemness and memory T cell formation include, amongst others, FOXO, Wnt/β-catenin and STAT3 [118,119]. Therefore, further research is required to better understand the contribution of glycolysis in the context of PI3K/Akt/mTOR blockade.

At the intersection between glycolysis and OXPHOS is pyruvate, which enters the mitochondria via the mitochondrial pyruvate carrier (MPC). A recent study showed that genetic MPC deletion favored a memory phenotype (through a preferential fueling of the TCA with fatty acids and glutamine) without affecting CD8^+^ T cell effector functions. Interestingly, MPC deletion did not improve CAR T cell effector functions in a nutrient-deprived TME. However, this approach remains interesting in view of the in vitro CAR T cell expansion. In fact, pre-treatment of CAR T cells with a small molecule inhibitor of MPC led to higher proportions of memory cells and a superior in vivo anti-tumor activity [120].

### 6.3. Promoting Mitochondrial Metabolism and Fitness

Given the fact that mitochondrial metabolism is linked to stemness and in vivo persistence of CAR T cells, several strategies (beyond the choice of co-stimulation) are currently being investigated to skew the CAR T cells’ bioenergetics accordingly. A key regulator of mitochondrial biogenesis is PGC1α. PGC1α is triggered by enhanced energetic demands, such as those imposed by T cell activation in the TME. It is activated by phosphorylation following AMPK activation and functions via the binding and activation of DNA-binding transcription factors, including nuclear respiratory factors, NRF1 and NRF2, which are involved in controlling the excess of ROS generated upon increased mitochondrial activity [121,122]. Several studies have addressed the impact PGC1α overexpression. Bengsch and colleagues reported that during chronic viral infection, T cells undergo exhaustion and are unable to meet their bioenergetic demands as both glycolysis and OXPHOS are impaired [55]. This “metabolic paralysis” was caused by mitochondrial depolarization, which did not allow the build-up of a proton gradient across the mitochondrial membrane. Membrane potential and function were restored by PGC1α overexpression. In another study, the same approach also led to increased mitochondrial biogenesis, and subsequently increased the frequency of memory CD8^+^ T cells [86]. Several studies have demonstrated the benefits of combining CAR T cells with an immune checkpoint blockade [7,8,9]. In fact, interfering with PD-L1/PD-1 interaction leads to a metabolic switch towards FAO and OXPHOS in T cells, thereby promoting cell survival and self-renewal [54]. Combining anti-PD-1 treatment with the PGC1α agonist benzafibrate in a murine melanoma model led to increased OXPHOS and reduced apoptosis in T cells [32]. Interestingly, in a different study, it was shown that the mitochondrial function of T cells represents a marker for responsiveness towards immune checkpoint blockade treatment [87]. As discussed above, 4-1BB co-stimulation shifts metabolism towards OXPHOS while promoting mitochondrial biogenesis and fusion [92].

Modulating mitochondrial dynamics (i.e., fusion or fission) represents an alternative strategy for bolstering mitochondrial fitness. This is achieved by pharmacological means, such as Mdivi-1, a mitochondrial division inhibitor, or ablation of genes such as dynamin-related GTPase (*DRP*1) or mitochondrial division dynamin I (*DNM*1) [89]. Both approaches led to fused mitochondria, increased mitochondrial fitness, and anti-tumor activity of the tumor-infiltrating lymphocytes.

Recent studies have exploited the potential of (over-)expressing metabolic enzymes to render CAR T cells more resilient towards the TME. Two candidate molecules are *Lactobacillus brevis* NADH oxidase (*LbNOX*) and D-2-hydroxyglutarate dehydrogenase (*D2HGDH*) [123], which catalyze the transfer of electrons from O_2_ to H_2_O_2_ and oxidation of D-2-hydroxyglutarate (D-2-HG) to alpha-ketoglutarate (αKG), respectively. Co-expressing *LbNOX* and CD28ζ CAR induced higher levels of baseline OCR as compared to the control group. In addition, these CAR T cells were not as affected by OXPHOS inhibitors such as antimycin A and rotenone. Overall, survivability was increased but not in vivo efficacy. D2HGDH-expressing CAR T cells metabolize D-2-HG, which is abundant within the TME. However, basal and maximal respiration is reduced, while the frequency of TME-infiltrating T_EM_ cells is increased. The latter could also explain the improved anti-tumor activity and emphasizes that the relationship between metabolism and function is not linear, but more complex. Kynurenine accumulation is also frequently found in the TME and results from an overexpression of indoleamine 2,3-dioxygenase 1 (IDO1) or tryptophan 2,3-dioxygenase 2 (TDO) in tumor and/or tumor-associated cells (e.g., MDSCs or cancer-associated fibroblasts). It has a strong (CAR) T cell-inhibitory effect (compared to others’ metabolic interference [124]) and can limit the efficacy of immune-based therapies [125,126]. CAR T cells which were modified to express the kynurenine-degrading kynureninase showed a better proliferative and tumor-eradicating capacity but glucose uptake was actually also increased. As described above, adenosine also negatively impacts T cell metabolic fitness and function. Therefore, an interesting approach is the expression of adenosine deaminase (ADA), which is a characteristic of CD26-expressing T cells [127]. It seems that ADA metabolizes adenosine, which is suppressive for T cells, into inosine, which can be used as an alternative carbon source under nutrient-deprived conditions. Its ribose subunit provides ATP and biosynthetic precursors from both glycolysis and the PPP. Besides supporting carbon metabolism, inosine has been proven to have boosting anti-tumor effects on effector T cells, even in the absence of glucose. For instance, inosine enhanced cytokine production (granzyme B, IFN-γ and TNF-α) and the tumor killing capacity of T cells, even in glucose-deprived conditions [128].

### 6.4. Nutritional Support

Given the competition over nutrients within the TME, several efforts are currently focusing on the potential supplementation of amino acids and nucleotides. For example, L-arginine can improve T cell anti-tumor activity both in vitro and in vivo, by enhancing OXPHOS and memory cell formation [129]. Regarding nucleotides, inosine seems to be an attractive supplement given its potential to enter glycolysis and the pentose phosphate pathway. For instance, in mouse models, it enhanced tumor clearance even under nutrient-deprived conditions. Similar to inosine, methionine can enter the carbon cycle and support T cell metabolic fitness and effector functions, mostly due to its role as methyl donor in nucleotide methylations and the epigenetic reprogramming needed for T cell differentiation. Furthermore, methionine is the amino acid required to start protein synthesis [124].

Other nutrients to investigate are fatty acids. As specified above, the accumulation of long-chain fatty acids is detrimental to T cells. However, short chain fatty acids (SCFAs), such as butyrate, propionate and acetate, seem to have the opposite effect. SCFAs are generated by bacteria present in the gut and are incorporated into T cells either via passive diffusion or binding to different transporters. How SCFAs affect T function depends on the cytokine milieu and T cell activation conditions. For instance, whereas CD4^+^ T cell activation in the presence of TGF-β1, IL-2 and 0,1 mM butyrate induces T_Reg_ generation [130], higher concentrations of butyrate, propionate and acetate support Th1 and Th17 differentiation [131]. In addition, CD8^+^ T cells cultured in the presence of 1 mM propionate and butyrate were shown to enhance IFN-γ and Granzyme B (GrzB) production. Interestingly, SCFAs were shown to promote long-lasting memory phenotypes through the upregulation of FoxO1, a transcription factor required for memory formation. This memory transition was supported by favoring the use of fatty acids and glutaminolysis to fuel the TCA rather than glycolysis [132,133]. In a different study, SCFAs could enhance CAR T cell anti-tumor activity via mTOR activation and the production of effector molecules, including TNF-α [134]. Therefore, depending on the concentration and culture conditions, SCFA can either favor a memory-like phenotype or induce the differentiation towards a specific effector T cell subset, both of which have their advantages, whereas memory-like T cells render the advantage of long-term responses and are required to establish an adaptive immune response [135], more differentiated effector T cells are essential to rapidly eliminate tumor cells [136,137]. Therefore, a balance between both types of T cells might provide the optimal result.

As mentioned above, the TME is characterized by a high concentration of ROS. Therefore, the supplementation of antioxidants might have a beneficial result in maintaining T cell fitness and function. The use of *N*-acetylcysteine (NAC) was proposed in a study to limit ROS metabolism during T cell activation [138]. It was shown that CD19 CAR T cells supplemented with NAC remained less differentiated (T_SCM_) and displayed overall reduced glycolytic metabolism, as evidenced by the reduced mTOR activity, lower expression of key glycolytic genes, such as Glyceraldehyde 3-phosphate dehydrogenase (*GAPDH*), Enolase 2 (*ENO*2), Pyruvate kinase (*PKM*) and Lactate dehydrogenase (*LDHA*), and reduced glucose uptake. In addition, NAC-treated cells showed increased expression of CPT1α, supporting FAO.

Self-evidently, metabolic re-modeling of the TME by interfering with the metabolic pathways and signaling in tumor and/or tumor-associated cells also represents an interesting alternative strategy for improving the environmental conditions for immune cells and is discussed in detail elsewhere [139,140]. 

## 7. Conclusions

Despite having revolutionized the field of cancer treatment, CAR T cell therapy still has several obstacles to overcome, including toxicity and efficacy. The latter is in most cases the result of successful tumor immune escape phenomena that impact the CAR T cells’ functionality and range from inadequate motility to reduced secretion of effector cytokines. In an attempt to overcome these limitations, the current focus is set on optimizing CAR T cell design with strategies such as including on/off switches to modulate CAR T cell activity and toxicity, CAR T cells redirected for universal cytokine-mediated killing, known as TRUCKs [141], universal CARs [142] or armored CAR T cells [143]. Currently ongoing clinical trials mostly focus on hematological malignancies. In fact, the combination regimens of CAR T cells and CPB are close to finding a cure to such diseases. On the other hand, intensive efforts are still ongoing in targeting several antigens expressed on solid tumors.

In addition to the currently developed CAR generations, significant effort is focused on optimizing CAR T cell culture, manufacturing and transfer with the aim of altering certain aspects of T cell fitness and metabolism. Emerging evidence emphasizes the interconnection between T cell metabolism and function, differentiation, and persistence. For this reason, it seems inevitable to evaluate metabolic modulation as a strategy to improve CAR T cell therapies. Easier-to-incorporate approaches deal with the optimization of culture conditions that can extend from the supplementation of cytokines and amino acids to the pharmacological control of metabolic pathways. The design of the CAR construct, especially the costimulatory domain, can strongly influence the metabolism, phenotype, persistence and possibly also toxicity. In the future, we expect a much stronger focus on the metabolic impact of the utilized signaling domains, which will at least co-determine CAR design. In addition, further genetic interventions are also conceivable which will allow certain key metabolic molecules to be more or less strongly expressed. Overall, these approaches focus on the generation of preferentially stem-like CAR T cells with high anti-tumor activity, superior survivability within the TME, and long-lasting persistence. In spite of all the possibilities and opportunities of such concepts, it should not be forgotten that we are talking about a combined cell-, immune-, and gene-therapy with corresponding regulatory requirements. Even small changes in manufacturing protocols may take a long time to reach clinical implementation (if they do at all). 

## Figures and Tables

**Figure 1 cancers-14-05442-f001:**
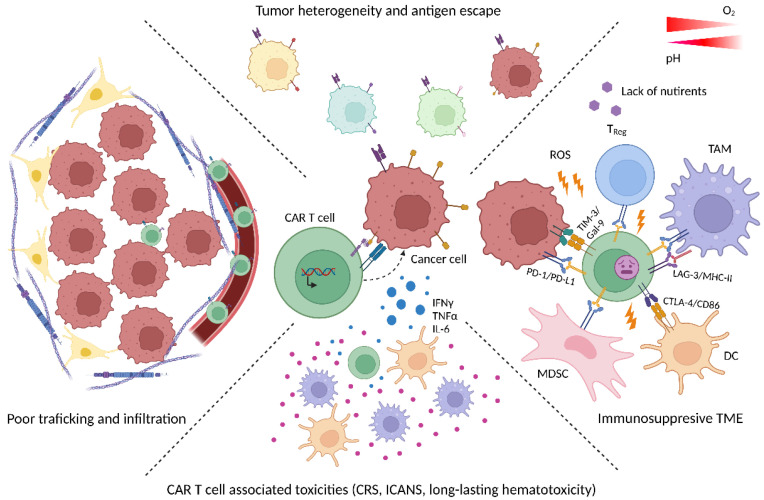
Current barriers in CAR T cell therapy. Clinical efficacy of CAR T cell therapy can be limited by various factors. The heterogeneity of tumors can give rise to variants that do not carry the target antigen on their surface. Immune checkpoint molecules (e.g., PD-L1 on tumor and/or stroma cells that binds to PD-1 on CAR T cells) can slow down anti-tumor activity. Overall, CAR T cells enter an immunosuppressive TME with tolerance-promoting cells such as myeloid-derived suppressor cells (MDSCs), tumor associated macrophages (TAM) or T_Regs_ and inhospitable metabolic conditions (i.e., hypoxia, acidosis, oxidative stress, depletion of critical nutrients, etc.). In fact, the TME and the tumor cells within can represent an impermeable obstacle for CAR T cells due to their poor trafficking resulting in an insufficient infiltration of tumor tissue. In addition, CAR T cells’ (over)activation can lead to severe and potentially life-threatening toxicities with cytokine release (CRS); immune effector cell-associated neurotoxicity syndrome (ICANS) and long-lasting hematotoxicity as the most common.

**Figure 2 cancers-14-05442-f002:**
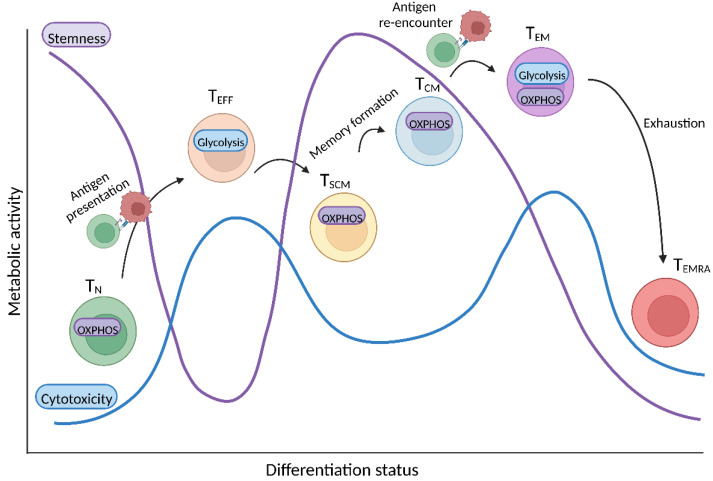
T cell differentiation is interconnected to metabolism. As naïve T cells (T_N_) leave the thymus and undergo antigen presentation, they experience a metabolic switch from oxidative phosphorylation (OXPHOS) to glycolysis as they become effector T cells (T_EFF_). After antigen clearance, a small subset of T cells become stem central (T_SCM_) and central memory (T_CM_) T cells, which mostly rely on OXPHOS. In case of TCR re-engagement, effector memory T cells (T_EM_) rapidly shift to a high glycolytic activity that supports their strong effector function. Lastly, terminally differentiated T cells (T_EMRA_) appeared rather senescent. T cell subsets that rely more on OXPHOS (T_N_, T_SCM_, T_CM_) are characterized by stemness, self-renewal and proliferation capacity. On the contrary, glycolysis-fueled cells (T_EM_, T_EFF_) display high cytotoxic function that ultimately leads them to senescence (T_EMRA_).

**Figure 3 cancers-14-05442-f003:**
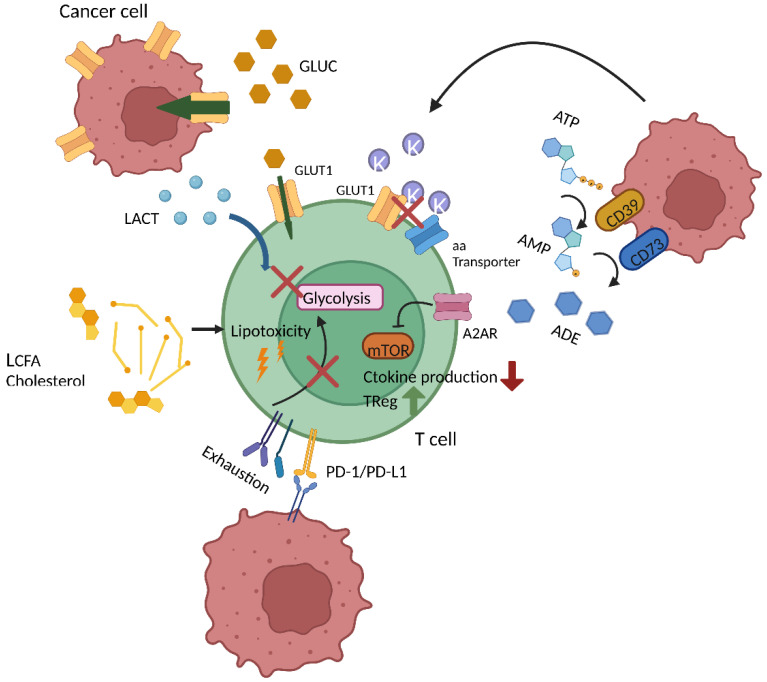
Metabolic barriers for T cells in the TME. The metabolic fitness of T cells can be compromised in a variety of ways within the TME. Critical nutrients such as glucose (GLUC) are depleted. Bioenergetically active tumors secrete bioactive molecules such as lactic acid (LACT), which impedes glycolysis. Dying cancer cells release vast amounts of potassium (K) that limits glucose and amino acid uptake. The two ectonucleotidases CD39/CD73 degrade ATP into adenosine (ADE), which also impairs metabolic T cell fitness. Increased levels of lipids together with an increased fatty acid uptake of TME-infiltrating T cells promotes exhaustion and lipotoxicity.

**Figure 4 cancers-14-05442-f004:**
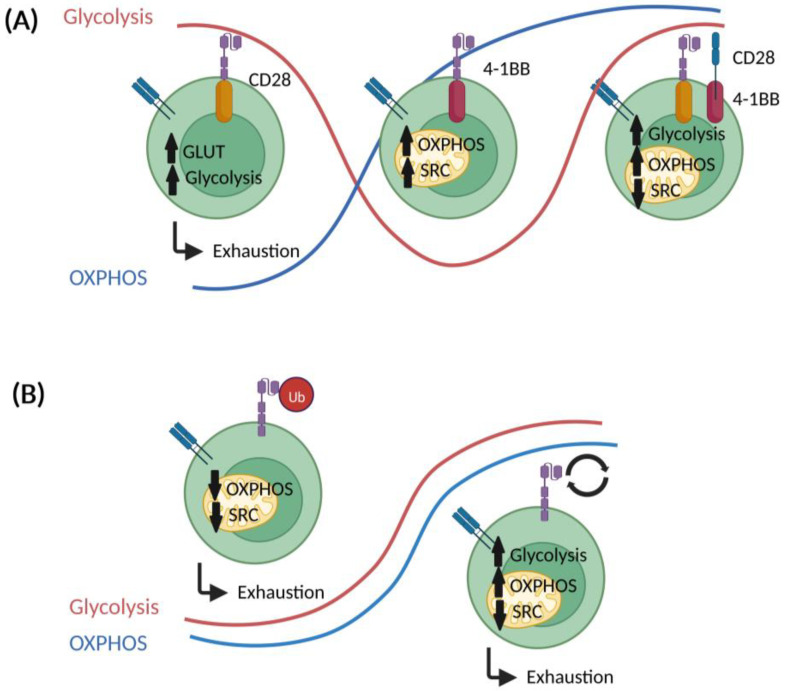
Impact of CAR design on CAR T cell metabolism. (**A**) Co-stimulatory signaling domains (CD28, 4-1BB, CD28+4-1BB) of CARs can impact several metabolic parameters such expression of glucose transporter (GLUT), extracellular acidification rate (ECAR), oxygen consumption rate (OCR) or spared respirator capacity (SRC). (**B**) Ubiquitination (Ub) and tonicity of CARs with impact on bioenergetics (and exhaustion).

**Figure 5 cancers-14-05442-f005:**
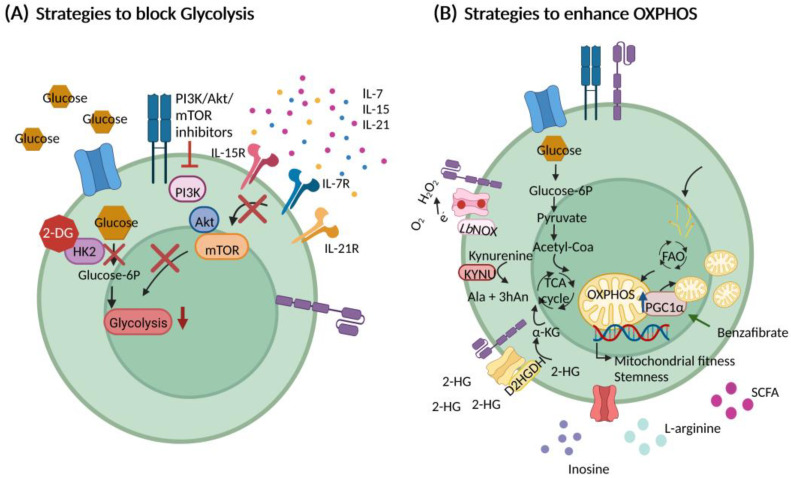
Strategies to modulate CAR T cell metabolism. (**A**) Strategies to block glycolysis. Cell culturing conditions can be improved by optimizing the use serum-containing and/or serum-free media. Treatment with cytokines such as IL-7, IL-15 or IL-21 led to the generation of the preferred T cell phenotype. Metabolic skewing (towards FAO and OXPHOS) can also be achieved by interfering with glycolysis by interfering with glycolytic enzymes such as hexokinase 2 (HK2) or glycolytic signaling such as the PI3K/Akt/mTOR axis. (**B**) Strategies to enhances oxidative phosphorylation (OXPHOS). Genetic engineering can be utilized for driving mitochondrial biogenesis and fitness (by, e.g., PGC1α) or metabolizing/redirect substrates such as 2-HG (by, e.g., D2HGDH), kynurenine (by, e.g., kynureninase) or O_2_ (by, e.g., *LbNOX*). Supplementation of nutrients such as inosine, L-arginine or short chain fatty acids (SCFAs) could also have a beneficial effect for CAR T cell metabolism and stemness. Abbreviations: Phosphoinositide 3-kinase (PI3K), protein kinase B (Akt), mammalian target of rapamycin (mTOR), interleukin (IL), interleukin receptor (IL-R), fatty acid oxidation (FAO), oxidative phosphorylation (OXPHOS), Peroxisome proliferator-activated receptor-gamma coactivator alpha (PGC1α), short chain fatty acids (SCFA), 2-Deoxy-D-glucose (2-DG), hexokinase 2 (HK2), glucose-6-phosphate (glucose-6P), tricarboxylic acid (TCA), alpha-ketoglutarate (α-KG), 2-hydroxyglutarate (2-HG), D-2-hydroxyglutarate dehydrogenase (D2HGDH), Kynureninase (KYNU), 3-hydroxyanthranilic acid (3hAn), alanine (Ala), *Lactobacillus brevis* NADH oxidase (*LbNOX*).

## Data Availability

Not applicable.

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
