# Peer review of "Role of CAR T Cell Metabolism for Therapeutic Efficacy"

_cancers, 2022, doi:10.3390/cancers14215442_

Round 1
Reviewer 1 Report
A very nice review on a timely topic. I only have some minor comment regarding the figures. While the symbols and colors look nice, I find it somewhat difficult to quickly get a sense of what the message is.
Specifically in Figure 2. It is not explained, what the vertical position means (higher meaning more active?). The metabolic lines blue and purple suggest a cell progressing through a fate, but this is not the case. Also, the symbols for cytotoxicity and stemness are used rather arbitrarily - how is defined one versus two versus three symbols?
Figure 3. Several aspects of the figures are unclear. E.g. what do the Kalium symbols mean? Are they leaving the cell or entering? What receptors are "blocked" in the presence of K? This figure not very novel, the aspect of CAR should be in focus.
Figure 4 is, in my view, is very important, as it addresses the topic of the manuscript. Maybe it could be made clearer, how metabolism changes (the reader might not understand ECAR OCR SCR quickly).
Figure 5. It is unclear, what is a proposed strategy for an intervention is and what is the known established metabolic pathway. This figure very ambitious, could be restructured to improve clarity.
Author Response
We thank Reviewer 1 for the helpful comments that helped us to improve our manuscript.
1. Figure 2 was modified including the axis labeling and the representation of stemness/cytotoxicity. The description of both parameters is based on literature research and they are suitable to compare the different T cell phenotypes with each other. Together with changes in the main text, the figure should be easier to contextualize.
2. As specified in the main text, calcium symbols represent (mostly tumor cell-derived) calcium ions present within the TME. High concentrations of calcium block glucose and amino acid transporters. This has been clarified in the updated figure.
Although this figure might not be very novel, it was included to explain how T cell metabolism is dampened in the tumor microenvironment as a basis to go deeper into CAR T cell metabolism later on in the manuscript.
3. Figure 4 was updated to make it clearer. ECAR/OCR was substituted by glycolysis/OXPHOS, what may be easier and faster for the reader to understand. In addition, curves that represent the metabolic phenotype more intuitively were added.
4. By dividing the figure into two parts, we have (so we believeI) been able to improve the clarity.
Reviewer 2 Report
Saborido et al provide a nice review of the role of metabolism in CAR T cell therapies. This is a useful topic with important basic science and clinical interest, but the following items still need to be addressed.
Major issues
1) Need to update Figure 1.
a) Have TAM listed under CAR T cell associated toxicities but in the text it is discussed as part of the immunosuppressive TME. Move that and include in the figure legend what TAM stands for.
b) Add long lasting hematotoxicity to the text of the CAR T cell associated toxicities portion of the figure and to the figure legend.
c) Add IL-6 to the CAR T cell associated toxicities figure (near IFNg, TNFa) since the text spends the most time discussing IL-6.
2) Include more details in the manuscript about why the metabolite methyl glyoxal paralyzes T cells (Lines 116-117): Thus, MDSCs paralyze T cells by transferring the metabolite methyl glyoxal while TRegs perform redox remodelling of TME.
3) Include more details in the manuscript about how Tregs are functioning in the TME in addition to redox remodeling of the TME (Lines 116-117). Thus, MDSCs paralyze T cells by transferring the metabolite methyl glyoxal while TRegs perform redox remodelling of TME.
4) Need to update Figure 2, it is currently hard to understand. Need an X and Y axis to better understand the figure or something to explain what is causing these cells to differentiate. Without some clarity on that you could just list the cell types and their cytotoxic, stemness, and metabolic profile.
5) Would be important to address if there have been combination clinical trials using CARs and checkpoint inhibitors and what has been learned from those (Lines 241-242 as well as Lines 417-419). What are the take home points from the studies cited?
6) Add more details to the manuscript on the function of PGC1a regulating mitochondrial biogenesis (Lines 409-410).
7) Add more details to the manuscript on the function of inosine supporting carbon metabolism and anti-tumor functions (Lines 454-456).
8) Need to discuss more in depth the advantages and disadvantages of moving from a stem cell phenotype to a specific T cell subtype such as Th17 in regards to CAR T cell therapy success in a tumor microenvironment (Lines 469-483).
9) Update Figure 5 to visualize strategies to promote oxidative phosphorylation (Lines 517-518).
10) Update the conclusion section to provide the reader with an idea of what strategies are currently working and what strategies have strong momentum at this time (Lines 536-555)
Minor issues
1) Update sentence for clarity and brevity (lines 23-25) - However, as of to date currently a substantial proportion of patients receiving CAR T cells will not reach long-term full remission.
2) Update sentence for clarity and brevity (lines 27-30) – Thus, T-stem cell memory (TSCM) cells, which have an enhanced capacity for self-renewal and persistence, preferentially meet their energetic demands through oxidative phosphorylation (OXPHOS) as opposed to the rather glycolytic effector T cells (TEFF).
3) Update sentence for clarity and brevity (lines 49-51) – CARs are synthetic constructs that bind to a specific target antigen in a major histocompatibility complex (MHC)-independent fashion. In the follow, and they trigger a vigorous T cell activation leading to the target cells’ elimination.
4) Update sentence for clarity and brevity (lines 108-109) – The TME is (also from the immunometabolic perspective) a hostile milieu for (CAR) T cells including from an immunometabolic perspective.
5) Update sentence for clarity and brevity (lines 109-111) – Critical (for T cell function) Nutrients critical for T cell function such as glucose or arginine are depleted and detrimental metabolites such as kynurenine, reactive oxygen species (ROS), and lactic acid 110 accumulate at the same time.
6) Update spelling on remodeling (Lines 116-117): Thus, MDSCs paralyze T cells by transferring the metabolite methyl-116 glyoxal while TRegs perform redox remodelling of TME.
Author Response
We thank Reviewer 2 for the helpful comments that helped us to improve our manuscript.
- Figure 1 was updated including all comments.
- Text was updated including more information on this topic.
- Text was updated including more information on this topic.
- Figure 2 was modified to clearer. X and Y axis were added, as well as some brief clarifications in terms of events that trigger phenotypic changes (i.e., antigen presentation, memory formation, antigen re-exposure, exhaustion etc.).
- Text has been updated including information on clinical trials using CARs in combination with CPB.
- Text has been updated with additional information regarding PGC1 a.
- Text has been updated with additional information regarding inosine.
- Text has been updated with additional information discussing the paradigm between memory and effector T cell subsets. Although it is already discussed in other parts of the manuscript.
- Figure 5 has been updated separating the figure into two parts. Part A concentrates on interfering glycolysis, Part B on promoting OXPHOS.
- Section "conclusion" was updated including current and future perspectives in the field of CAR T cell therapy, mostly focusing on new generations of CAR T cells constructs, combinatorial treatments with CPB, and treatment of solid tumors.
- All minor points were addressed.
Reviewer 3 Report
In this manuscript, Authors describe role of CAR T cell metabolism for therapeutic efficacy. Authors present both limitations and advantages of CAR T cell therapy. They suggest to inevitable to evaluate metabolic modulation as a strategy to improve CAR T cell therapies. This paper is well written and valuable. The paper is topical and very important for the improvement of CAR T therapy.
I propose that in section 2, section 5 and section 6, sub-headings were numbered, e.g. 2.1 ; 2.2 ….and be in italics. These sections will be clearer.
Author Response
We thank Reviewer 3 for the helpful comments that helped us to improve our manuscript.
1. Subheadings were added to make the separation between sections clearer.
Reviewer 4 Report
In this manuscript, the authors made a comprehensive review on the roles of CAR T cell metabolism in modulating therapeutic efficacy and discussed possible strategies to improve CAR T therapy. Overall, the manuscript is well written, and the illustrations are delicately designed. The merit of this article may be augmented by language editing from a native English speaker. For example, the authors may find a better way of expression for the sentences in line 23 “However, as of to date…” and in line 48 “resulting in several approvals by the FDA”.
Author Response
We thank Reviewer 4 for the helpful comments that helped us to improve our manuscript.
1. Minor spell check was performed.
Round 2
Reviewer 2 Report
Looks good.